# Chronic Pancytopenia Due to Centrally Mediated Hypothermia in Two Children with Severe Neurological Impairment

**DOI:** 10.3390/children7040031

**Published:** 2020-04-08

**Authors:** Karen Fratantoni, Julie M. Hauer

**Affiliations:** 1Division of General and Community Pediatrics, Children’s National Hospital, Washington, DC 20110, USA; KFratant@childrensnational.org; 2Division of General Pediatrics, Boston Children’s Hospital, Boston, MA 02115, USA; 3Seven Hills Pediatric Center, Groton, MA 01450, USA

**Keywords:** neurological impairment, cerebral palsy, neurodisability, hypothermia, thrombocytopenia, pancytopenia, children

## Abstract

We report on recurrent pancytopenia over five years in two children with severe impairment of the central nervous system. Assessment by hematology did not identify an etiology, including bone marrow biopsy in one. Both patients had sustained normalized blood cell counts following interventions to maintain or return to a temperature above 33 °C. Acute cytopenias following medically induced and environmental hypothermia have been reported. Recurrent pancytopenia due to centrally mediated hypothermia in patients with severe neurological impairment is often not recognized, putting such children at risk for unnecessary testing and transfusions. We provide a practical approach to management that is feasible for caregivers in the home setting with suggestions for monitoring.

Children with severe impairment of the central nervous system (CNS), often referred to as severe neurological impairment (SNI), are at risk for recurrent hypothermia as a result of hypothalamic dysfunction. SNI has been defined to indicate a group of children with CNS disorders, resulting in both motor and cognitive impairment as well as medical complexity. These children require assistance with most or all of their activities of daily living. The condition is permanent but can either be static or progressive [1]. Hypothermia is defined as a temperature less than 35 °C. Acute cytopenia following hypothermia from environmental exposure or induced as medical therapy has been reported [2,3,4]. This is the first report of chronic pancytopenia, indicating a reduction in all three blood cell lines, due to centrally mediated hypothermia in two patients with SNI. Evaluation for an etiology was negative, including a bone marrow biopsy in one. Both received blood transfusions. All blood cell counts normalized following efforts to maintain or return to a temperature above 33 °C. Practical monitoring and management strategies that are feasible in the home setting are provided. Recognition of this risk can minimize unnecessary tests and interventions. 

Parents/legal guardians gave written informed consent. This report was approved by the local research ethics board. This case report was structured as per the case report guidelines [5]. 

## 1. Case Report

### 1.1. Case 1

A 20-year-old male with SNI has recurrent pancytopenia for five years. His past medical history includes traumatic brain injury early in life, resulting in spastic quadriparesis, severe intellectual disability, visual impairment, and seizure disorder. Other notable details include tracheostomy, overnight ventilation, acute pancreatitis, cholecystectomy, and nephrolithiasis with external shockwave lithotripsy. He receives all feeds, fluids, and medications by gastrostomy feeding tube. He was admitted to a long-term care facility (LTCF) in May 2018. Prior to admission, the patient had recurrent pancytopenia without clear etiology; he was evaluated by hematology, and a bone marrow biopsy was unremarkable. He received multiple transfusions January 2015 through April 2017. The facility’s hypothermia care plan was implemented, which includes daily temperature monitoring and interventions as noted later for temperatures below 35 °C. His temperature since admission has commonly been 33.8 °C to 35.5 °C. Temperatures below 33.5 °C are noted to take longer to return to the desired range, typically within eight to 12 h. Table 1 includes the data obtained from emergency department and clinic visits prior to admission to LTCF, as well as blood cell counts and temperature after admission. All blood cell counts normalize with a temperature greater than 34 °C as noted in Table 1.

### 1.2. Case 2

A 22-year-old male with SNI has recurrent pancytopenia for five years. His past medical history includes a severe hypoxic event at birth due to placental abruption. Resulting problems include spastic quadriparesis, severe intellectual disability, seizure disorder, cortical visual impairment, gastrostomy feeding tube, tracheostomy, and overnight ventilation. Valproic acid was discontinued in January 2016 without sustained benefit in blood cell counts. A hematology consultation did not identify any further etiology for his recurrent pancytopenia. Parents declined a bone marrow biopsy. Interventions included one transfusion in 2018. His temperature on admission for two respite stays was less than 33.3 °C each time. He was then admitted for long-term care in June 2019. Similar to case 1, use of the hypothermia protocol at the LTCF has maintained or resulted in a slow return of his temperature to the desired range, with improvement in all blood cell counts (Table 2).

## 2. Discussion

Hypothermia is used as a therapeutic intervention to reduce ischemic injury following neonatal hypoxic-ischemic encephalopathy, post cardiac arrest encephalopathy, and traumatic brain injury [2,3,4]. Acute causes of hypothermia include accidental hypothermia due to environmental exposure or cold-water immersion. Other causes include metabolic, drug induced, and sepsis. 

In contrast to acute causes, hypothermia can be chronic due to hypothalamic dysfunction [6]. The preoptic area of the hypothalamus is responsible for thermoregulation. Normally, a drop in body temperature is detected by peripheral temperature receptors, resulting in skin vasoconstriction and contraction of the erector pili muscles to reduce heat radiation, shivering to produce heat, and sympathetic excitation and thyroxine production to increase the basal metabolic rate [6]. Injury and structural changes in this area of the hypothalamus can alter this regulation. Shapiro syndrome is a rare disorder consisting of paroxysmal hypothermia, hyperhydrosis and agenesis of the corpus callosum. There is variability in the duration of the episodes, and the length of time between episodes can vary from months to years [5]. This syndrome may be due to a neurotransmitter disorder [7].

Cytopenias, most commonly thrombocytopenia, have been reported with hypothermia. Most case reports are of acute hypothermia following medically induced and environmental hypothermia [4,6,8,9]. Case reports of hypothermia and cytopenias have been reported with Shapiro syndrome, without the neurodisability described in our two cases [6,10]. Pancytopenia as a result of hypothermia following resection of a craniopharyngioma was reported in two cases, with no identified etiology other than chronic hypothermia [11,12]. Recurrent thrombocytopenia with hypothermia was reported in two children with severe disabilities from neonatal infection (congenital herpes encephalitis and Escherichia coli meningitis), with resolution each time upon rewarming [13]. Our two cases are the first to report recurrent pancytopenia in children with SNI and centrally mediated hypothermia. In addition to cytopenias, acute pancreatitis [8,12,14] is another reported effect of hypothermia, including in children with SNI [15].

Potential etiologies of hypothermia-induced cytopenias include liver and splenic sequestration, white cell marginalization, and possible marrow suppression [11]. Neutropenia related to low body temperature in hibernating animals is due to temporary margination, rather than decreased production or apoptosis, and is thought to eliminate thrombotic risk during periods of reduced metabolic activity [16,17].

## 3. Management and Practical Considerations

Improved temperature regulation has been described with clonidine and cyproheptadine in Shapiro syndrome [6], suggesting these medications may be beneficial in those with a neurotransmitter disorder and typical neurodevelopment. In contrast, hypothermia in children with global impairment of the CNS may be refractory to medications [6]. 

Risk from hypothermia-induced thrombocytopenia may be low in children with SNI, with no documented platelet count below 20,000 and low risk of bleeding with injury in non-ambulatory patients. Cytopenias are uncommon above 34 °C and resolve upon rewarming [10,11,12,15]. This information and temperature data from the LTCF suggest that significant cytopenias, along with other effects such as acute pancreatitis, occur when there is a sustained temperature below 33 °C for longer than eight to 12 h. This information can guide practical considerations for home care plans that are feasible and safe, while avoiding excessive monitoring and interventions. We suggest the following approaches to care:

Monitor temperature twice daily and as needed. If below 35 °C (95 °F) apply blankets and a hat, and consider brief use of heating pad.

Consider routine use of a hat and scarf to minimize scalp heat loss along with other layers in those with daily hypothermia.

Adjust the plan to the needs of the child. As an example, 34 °C to 35 °C is a general guideline given that it is easier to maintain a temperature than to raise it once it is below 33 °C. This can be individualized by recognizing reports of distress when a temperature is artificially too high in some individuals, reported at temperatures of 34 °C to 35 °C [13]. 

Care plans at the LTCF are also individualized so as to not interfere with attendance at school and activities. This is done by using extra layers overnight to artificially start with a higher temperature in the morning, allowing a lower temperature during the day, and not checking the temperature while in school. 

If cytopenia is noted in a child with SNI and hypothermia, when at typical baseline, it can be reasonable to monitor the child while using interventions to rewarm before rechecking for improvement in blood cell counts. When extra layers are applied for a temperature below 33 °C (91.4 °F), documentation at the LTCF demonstrates that it can take up to eight hours or longer for the temperature to increase to a desired range.

## 4. Conclusions

Hypothermia as a result of altered hypothalamic temperature control is a plausible explanation for recurrent cytopenias in children with SNI. These two patients highlight the benefit of care plans that minimize sustained hypothermia. The data suggest that risk occurs when a temperature remains below 33 °C for more than 12 h. Parents can be provided with care plans that are feasible to utilize in the home setting and individualized as needed. Monitoring for resolution of cytopenias upon rewarming minimizes testing and lowers cost. These cases, along with the benign nature of minimizing a sustained temperature below 33 °C, illustrate the benefit for such children. 

## Figures and Tables

**Table 1 children-07-00031-t001:** Case 1 temperatures and blood cell counts.

	3/28/16	9/28/16	11/28/16	1/11/17	3/20/17	5/20/18	9/27/18	5/14/19	10/13/19
Location	ED	Clinic	ED	ED	ED	LTCF	LTCF	LTCF	ED
Temperature (Celsius)	31.4	NA	30.3	36.3	30.2	33.6	35.9	34.2	35.9
Platelet	28,000	63,000	21,000	125,000	53,000	80,000	127,000	159,000	142,000
Hemoglobin	8.0	10.3	8.4	11.7	8.8	14.7	13.1	13.2	12.8
WBC	3120	4660	2300	4320	2910	4300	13,000	5000	15,000 *

ED = emergency department, LTCF = long term care facility, NA = not available, WBC = white blood cell count. * Prednisone dose six hours prior to WBC results. Lab value normal range: Platelet 146,000–326,000; Hemoglobin 12.5–16.0; WBC 5200–10,600.

**Table 2 children-07-00031-t002:** Case 2 temperatures and blood cell counts.

	12/28/15	1/13/16	2/3/16	3/25/16	3/18/16	4/7/16	3/31/17	7/11/19	1/23/20
Location	Clinic	Clinic	Clinic	Clinic	ED	Clinic	ED	LTCF	LTCF
Temperature (Celsius)	NA *	34.8	36.3	NA	34	NA	34.5	35.2	35.4
Platelet	47,000	104,000	542,000	71,000	55,000	59,000	90,000	153,000	250,000
Hemoglobin	6.9	6.9	8.0	9.7	10.6	8.1	12.1	14.4	12.7
WBC	3370	2970	7110	3650	3430	2850	5290	7300	6800

ED = emergency department, LTCF = long term care facility, NA = not available, WBC = white blood cell count. * Temperature typically less than 35 °C, often less than 33 °C, prior to admit to LTCF per parent report. Lab value normal range: Platelet 146,000–326,000; Hemoglobin 12.5–16.0; WBC 5200–10,600.

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
