# Peer review of "Chronic Pancytopenia Due to Centrally Mediated Hypothermia in Two Children with Severe Neurological Impairment"

_children, 2020, doi:10.3390/children7040031_

Round 1
Reviewer 1 Report
Dear Author
your manuscript has been carefully evaluated. I found it interesting.
I think that the manuscript is interesting but need substantial changes.
In specific:
- Definition of "severe neurological impairment". Definition of "pancytopenia". Add related bibliographic data.
- Case 1: The patients has a complex clinical picture, there is a multi organ involvement with respect to which no etiological, therapeutic data is provided nor any temporal link is expressed between the onset of pancytopenia, the other pathologies described and the supposed connection with the involvement of the nervous system and its several impairment.
- Case 2: Even for case 2, the data relating to a correlation between pancytopenia, severe neurological impairment and hypothermia are unclear and do not seem to have sufficient scientific value, however they have an informative value.
- Specify how it was established to apply hypothermic treatment and the timing of therapy, illustrate the therapeutic protocol applied with any pre-treatment and control investigations.
-Implement bibliographic data by carefully reviewing the literature related to hypothermia
Author Response
Reviewer 1
Thank you for your invaluable input. To improve our manuscript, we made the following changes / additions in your response to this input:
- Definitions for severe neurological impairment and pancyptopenia were added
- The risk for hypothermia due to hypothalamic dysfunction is part of the introduction
- We have added pertinent clinical findings to better describe the cases
- We have added more information on the facility’s hypothermia plan
Reviewer 2 Report
The manuscript by Fratantoni and Hauer reports 2 cases of children with severe central nerves system impairment, that causes recurrent hypothermia as a result of hypothalamic dysfunction. Hypothermia is defined as temperature less than 350C. One of the periphery phenotypes of these children is chronic pancytopenia. Upon close inspection including one bone marrow biopsy, hematopoiesis per se does not seem to be the cause for pancytopenia. Rather, once the body temperature is warmed up above 33oC, patients’ blood cell counts can be normalized.
The authors argue that patients with pancytopenia induced by hypothermia due to severe neurological impairment are often subject to unnecessary testing and blood transfusion. The manuscript discussed a practical approach to feasible management for caregivers in home setting with suggestions for monitoring.
I think it is significant that this is the first report aiming to managing such said conditions, and the management is feasible that helps to avoid unnecessary medical procedures. The cases are clearly presented longitudinally. However, I would like to see normal values included as a point of comparison either incorporated in the table or clearly stated in the text.
Author Response
Reviewer 2
Thank you for the time taken to review this manuscript. We have now added normal values for all blood cell lines.
Round 2
Reviewer 1 Report
Dear Author
your manuscript has been carefully evaluated. I found it interesting in its field.
Both the description of the clinical cases and the implementation of the bibliography make the manuscript more complete and clearer the information provided.